# Naples Yellow Revisited: Insights into Trades and Use in 17th-Century Sicily from the Macro X-ray Fluorescence Scanning of Matthias Stomer's 'The Mocking of Christ'

Michela Botticelli [1,2,*], Costanza Miliani [3], Eva Luna Ravan [1,4], Claudia Caliri [1,2,*] and Francesco Paolo Romano [1,2]

1   Consiglio Nazionale delle Ricerche, Istituto di Scienze del Patrimonio Culturale (CNR-ISPC), Via Biblioteca, 4, 95124 Catania, Italy; evaluna.ravan@uniroma1.it (E.L.R.); francescopaolo.romano@cnr.it (F.P.R.)
2   Istituto Nazionale di Fisica Nucleare, Laboratori Nazionali del Sud (INFN-LNS), Via Santa Sofia 62, 95123 Catania, Italy
3   Consiglio Nazionale delle Ricerche, Istituto di Scienze del Patrimonio Culturale (CNR-ISPC), Via Cardinale G. Sanfelice, 8, 80134 Naples, Italy; costanza.miliani@cnr.it
4   Department of Classics, Sapienza University, P. le A. Moro, 5, 00185 Rome, Italy
*   Correspondence: michela.botticelli@cnr.it (M.B.); claudia.caliri@cnr.it (C.C.)

**Abstract:** In a recent non-destructive analytical campaign at Museo Civico, Castello Ursino, in Catania, Italy, several paintings in the permanent collection were investigated by MA-XRF scanning, with a special focus on Matthias Stomer's production. On one depiction of the *Mocking of Christ* (ca. 1640) donated to the municipality of Catania by G.B. Finocchiaro in 1826, the analysis documented the use of Naples yellow. Sb with Pb was detected in yellow areas of the *Mocking of Christ*, but not in his work *Tobias healing his father*. This finding possibly suggested an early use of lead antimonate yellow in South Italy, although it is generally accepted that this pigment was introduced in painting in the eighteenth century. Further details on his technique and later conservation treatments are provided, as well as literary comparisons with the artistic production during the same period, in Sicily and elsewhere. A systematic study of Stomer's works, for example examining paintings produced while he was in Naples or Rome, might determine whether this material choice depended on local availability. Overall, it would shed light on his technique, as well as on the history of Naples yellow in southern Italy and beyond, before this pigment became so popular in the eighteenth century.

**Keywords:** MA-XRF; imaging; paintings; pigments; Naples yellow; lead antimonate yellow; Matthias Stomer

## 1. Introduction

A recent analytical campaign at Museo Civico, Castello Ursino, in Catania, Italy, has given the chance to study several paintings in its permanent collection [1], with a special focus on Matthias Stomer's production and its materiality [2]. This paper mainly focuses on the results of the non-invasive macro X-ray fluorescence (MA-XRF) scanning carried out on the *Mocking of Christ*. This type of investigation allowed real-time and remote visualisation of the distribution of chemical elements in the painting, in view of the description of original materials used by the artist, compositional changes at the production stage (pentimenti) and later conservation treatments. Particularly, the copresence of lead and antimony in the same areas, which appear to have been painted with a yellow colour, led to the hypothesis that Stomer knew and used Naples yellow for this painting.

### 1.1. Lead Antimonate Yellow through the Ages

In the early seventeenth-century manuscript by Mariani da Pesaro, a miniaturist, two types of potters' yellow are mentioned: one described as more subtle and good for colouring drawings and another with more body, both produced using antimony [3]. The

first, sometimes referred to as 'Type I', could be obtained from burnt lead, a mineral source of antimony and plain salt. For the second (Type II), salt was replaced by calcined wine residue, and a higher firing temperature was required. A third variety (Type III) was produced using Alexandrine tutty (also mentioned as *tuccia* or *tutia allessandrina*), which has been interpreted as a tin or zinc source alternatively [3].

Chemically, three different lead-based yellows exist. One which contains lead and tin, one with lead and antimony and one made of all these elements. In 1993, Khün [4] clarified that the first category (Pb + Sn) exists in two distinct crystalline forms, Type I and II. Although both are frequently mentioned with the general term *giallolino*, Type I might correspond to a pigment imported in Italy from abroad, as testified by common names such as *giallo Tedeschi*, *giallo di fiandra* or *luteolum Belgium*. Conversely, Type II is likely to stand for a pigment with earlier and discontinuous history—it begins to appear in fourteenth-to-fifteenth-century paintings and reappears in the sixteenth century—made in Venice under the name of *giallolino fino* and *giallo di vetro*, respectively. It has also been reported that 'A comparable recipe producing a lead tin-based yellow under the heading 'massicot' is to be found in a sixteenth century Flemish text, confirming the equivalence of the two terms' [5], although the term massicot more commonly describes an orthorhombic lead oxide, PbO, produced by heating lead white and used as a yellow pigment.

Roy and Berrie [6] have linked the manufacturing of *giallolino* to Venetian glass making, where lead and tin were common opacifiers and colourants. In the same technology, the second category (Pb + Sb) of yellow is a common product, in the form of yellow enamel pigment for the design of coloured maiolica or tin-glazed earthenware [7], a market that flourished in the fifteenth century all around Italy. For the third category of Pb + Sn + Sb yellows, in 1998 Roy and Berrie reported the use of a Pb–Sn antimonate phase in mid-seventeenth-century Roman paintings by Poussin, Gentileschi, Sassoferato and others [6], suggesting this yellow dates to around 1650 and has some connection to the area of Rome. However, Dik et al. [3] rejected this hypothesis based on the different origins (Pesaro or Venice, in northern Italy) of the manuscripts where recipes could be found at that time. Such ternary oxide yellows have been documented in seventeenth-century Italian paintings [8] and mid-European paintings of the eighteenth and nineteenth centuries [9]. It is not infrequent to find the ternary oxide or lead–tin yellows alternatively in admixtures with antimony yellow, one of the most remarkable examples being Raffaello's *Loggia di Amore e Psiche* in Rome, where all three types of yellow were used between 1517 and 1518 [10].

The occurrence and manufacture of antimony-based yellows in time and space, at least in glass making, can be established from several historical Italian treatises, where recipes cover a period spanning from the sixteenth century to the seventeenth century:

- 1456–1526: The *Calabranci Code* contains six different recipes for the preparation of yellow ceramic pigments using both Pb and Sb compounds. It is later included in Dionigi Marmi's *Segreti di furnace* (seventeenth century, Montelupo, Tuscany) [11];
- 1540: Vanoccio Biringuccio's *Pirotechnia* (Venice) provides the first evidence of the use of antimony for the manufacture of yellow pigments for pottery making, particularly for enamels and glass [6];
- 1548: Cipriano Piccolpasso, a potter from Casteldurante—now Urbania (the Marches), gave the first recipes for yellows from lead and antimony related to the main centres of glazed ceramics in the area comprised within current Tuscany, Umbria and the Marches;
- 1644: the first recipes for yellows containing lead, tin and antimony are found in the *Ricettario* Darduin of Murano (Venice), *Secreti per far lo smalto et vetri colorati*, with sources from 1523 [12].

In a systematic study on the occurrence of lead yellows in 220 paintings from the sixteenth and seventeenth centuries [13,14] it has been reported that, although Pb-Sn products are still the most common, the use of antimonate yellow becomes more frequent (44%, while 32% also contains tin). A less recent report [15] has shown that an increasing use of lead antimonate yellow characterises a wider artistic production (seventeenth century

to the twentieth century), the greatest popularity being between 1750 and 1850, while lead–tin yellow was losing favour until it finally disappeared from artists' palette at the beginning of the eighteenth century. It is not by coincidence that the first official recipe for the synthesis of the pigment lead antimonate yellow was published in 1766, in opposition to the natural version of *giallorino* growing in the proximity of Vesuvio, as mentioned by Cennino Cennini [16].

The rate of occurrence of antimonate yellow in paintings has been analysed by several authors and exhaustively summarised by Wainwright et al. [7], who highlighted an increasing rate of occurrence in European paintings from the seventeenth century to the eighteenth century and an inverted trend for the following century, its use also being to obtain shades of green, for example, by mixing it with azurite, Prussian blue or cobalt blue.

Despite the climax of its popularity in paintings matching the date of diffusion of the synthetic recipe, it is reasonable to hypothesise an earlier and less regular experimentation by painters, at least for those who had access to raw materials and were occasionally in contact with glass-making workers, because they operated in contexts where this technology was well developed, or other technologies had yet to come [17]. In fact, lead antimonate yellow has been identified in Lorenzo Lotto's *A Maiden's Dream*, ca. 1505 (on panel, Samuel H. Kress Collection, National Gallery of Art, Washington) and in the *Portrait of Giovanni della Volta and his Family*(?), probably 1547 (on canvas, National Gallery, London, NG1047) [6], with this evidence already setting the date of its introduction at about a century earlier than commonly thought.

A systematic Investigation of paintings dated before 1766 might produce a more comprehensive chronology of lead antimonate yellow in European paintings, whether it happened at different times and depending on where artists had their workshops.

*1.2. Matthias Stomer: The Sicilian Period*

After his youth in the Netherlands, where Matthias Stomer was born in 1600 and received training, becoming familiar with the old mannerists and Rubens' avant-garde [18], the painter travelled to the Italian peninsula [19]. Of the period he spent in Italy, three different phases are known: in chronological order, the Roman, Neapolitan and Sicilian periods. In Rome, Stomer arrived around 1630, when even the last *Caravaggista* had died and a classical reinterpretation of Caravaggio and baroque style were both in favour among Roman artists [20]. Zalapì [18] places the Neapolitan period between 1633 and 1637, although Nicolson had earlier referred to some uncertainty on the year he arrived in Naples [21]. According to the same author, it is also unclear whether Stomer first settled in Palermo or Messina after leaving Naples. However, documents prove that he was in Palermo in 1640 and remained in Sicily for several years [18,20] before going back to Rome. It is not known whether he died there or in Sicily around the middle of the century. The Sicilian production represents a good part of his career, with more than twenty of his paintings currently disseminated in several provinces, in major cities like Palermo, Messina and Catania, as well as in small villages like Linguaglossa (CT) or Caccamo (PA).

Such a wide diffusion is not only the result of Stomer's activity in different areas of Sicily but also of the regained popularity and recognition that the seventeenth-century artistic production experienced in the 1800s. Local wealthy families started to collect paintings from this period attributed to the Flemish, Neapolitan and Sicilian schools. Mancuso [22] explains this appreciation either with personal interest or emulation, as well as a collectors' market trend yet to be understood. No matter what the justification behind it is, there is clear evidence of a common tendency to collect paintings produced by painters active in Sicily or Naples during that period. Several examples can be found in Palermo, specifically, the Prince of Campofranco (Antonio Lucchesi Palli) and the Agostino Gallo collections, which include paintings by Domenichino, Caravaggio, Ribera, Van Dyck and Stomer. If these express the aristocratic intention of collecting valuable artworks for their own pleasure, the Finocchiaro collection, which includes the *Mocking of Christ* studied in

the present paper, documents the social aspirations of the emerging entrepreneur class and the great influence of coeval Palermitan collections.

## 2. Materials and Methods

### 2.1. The Mocking of Christ

The subject of this research is one of the two paintings of the *Mocking of Christ* by Stomer in the inventory of the Finocchiaro Collection, donated by Giovanni Battista Finocchiaro, a lawyer born in Catania and active in Palermo, to his native municipality in 1826. The 123 paintings in the collection, reasonably bought over the years in Palermo and finally shipped to the town hall in Catania, were transferred to the Benedictine Monastery of 'San Nicolò l'Arena' in 1874 before reaching their current location in Castello Ursino [23]. Both paintings are mentioned in the inventory as 'Cristo alla Canna' and described 'with lots of figures, Gherardo Della Notte's (alias Gerrit van Honthorst) School' [22]. The painting under investigation is oil on canvas (210 × 154 cm$^2$), painted around 1640.

All Stomer's artworks in the Finocchiaro Collection—including the *Catania Death of Cato*, the *Catania Death of Seneca* and the *Tobias healing his father*, the latter being mentioned in [21] at the time when it was still at Museo dei Benedettini—are attributed to the Sicilian period. Representing all indoor settings, the painter had deliberately chosen to transform the Caravaggisti concept of light, giving even more splendour and pathos to artificial lighting, mediated through the eyes of his Utrecht master van Honthorst.

In the frame of a large-scale analytical campaign to characterise the technique of fifteenth- and sixteenth-century master painters in the permanent collection of Museo Civico, Castello Ursino in Catania, paintings by Antonello de Saliba, Luis de Morales, Jusepe de Ribera and Matthias Stomer have been analysed using the innovative MA-XRF imaging system developed by the XRAYLab of ISPC-CNR, presenting real-time imaging capabilities and allowing the analysis of large areas in a fast scanning modality. The aim of the investigation was the identification of original materials, pentimenti and later interventions by imaging techniques [2]. For Stomer, the campaign has not only offered the opportunity to accomplish these goals. Beyond expectations, it has also given insights into the technical novelties of Stomer's art.

### 2.2. Macro-XRF Scanning

The investigation of the painting in this study has been carried out using a mobile MA-XRF scanner based on real-time technology designed and built at XRAYLab of ISPC-CNR in collaboration with the LANDIS group of the LNS-INFN in Catania [24]. The instrument is fully modular, and it is based on a low-power X-ray source focused with a polycapillary and two SDD detectors (50 mm$^2$ active area and 140 eV energy resolution at 5.9 keV) operated in parallel. An interchangeable Rh or Cr target can be selectively used as X-ray source, allowing the efficient excitation of both low and high Z elements and the investigation of some elements in the pictorial layer with a different analytical depth. The scanner is based on a three-axis system (XYZ) presenting a 110 × 70 × 20 cm$^3$ travel range. The scanning is performed in the vertical direction (XY), while the Z axis is used for alignment and offers the possibility of keeping the measurement head-painting distance constant thanks to a dynamic correction performed with a laser sensor.

The MA-XRF scanning is performed by positioning samples out of the polycapillary focus with the primary beam presenting a spot size of several hundred microns. The full area is covered in 4.3 h with a pixel size of 500 μm and 5 ms dwell time (i.e., 100 mm/s scanning speed). A lateral resolution down to 25 μm can be achieved at the focus position, allowing the use of the scanner for a high-resolution micro-XRF mapping of pigment materials. XRF spectra are processed on the fly using in-house software programmed under a real-time LabVIEW environment. It applies a least square fitting procedure of pixel XRF spectra and provides deconvoluted elemental distribution images in a live mode during scanning [24]. Additionally, several processing functions (i.e., RGB correlation maps,

scatterplots, PCA/NNMF, integral and maximum pixel spectrum) can be applied to the forming images, supporting data processing, elaboration and interpretation.

Six different areas were scanned on the *Mocking of Christ* (Figure 1) using the Rh target (high-energy mode, HE), with pixel size of 1000 × 1000 μm$^2$ and dwell time per pixel of 15 ms.

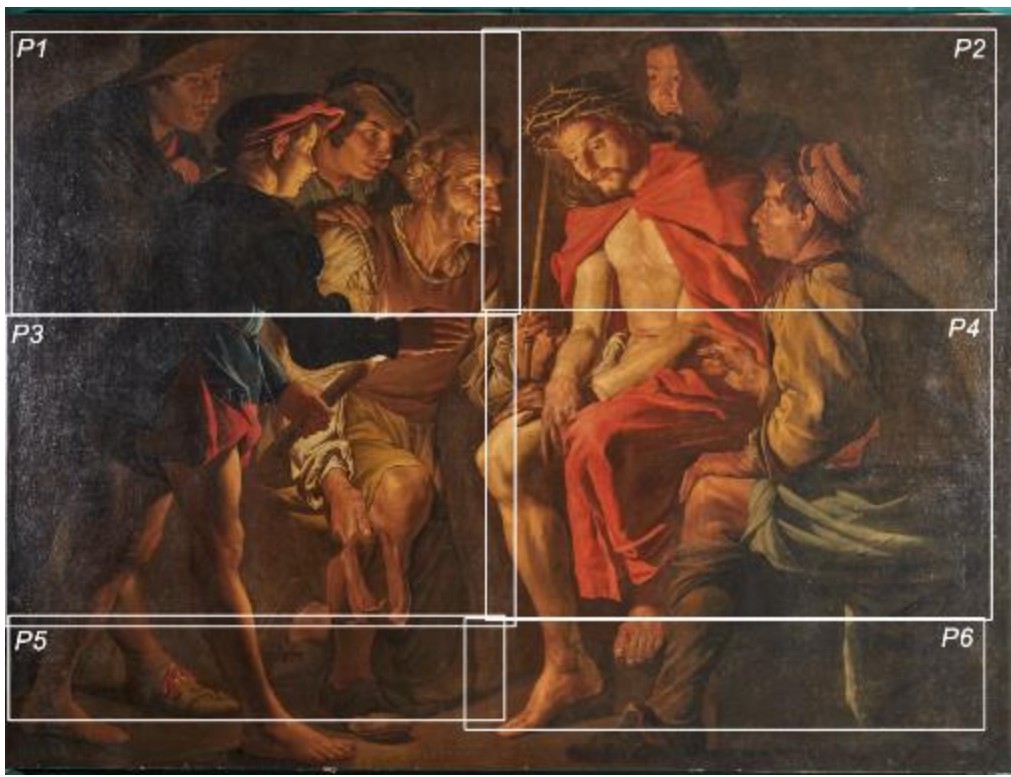

**Figure 1.** Visible light image of the *Mocking of Christ* by Matthias Stomer, in the permanent collection of Museo Civico, Castello Ursino, in Catania, Italy; rectangles and labels P1–P6 show the different areas scanned with the MA-XRF system.

Additionally, some details of the painting, i.e., the area around Christ's face, were scanned with the Cr target (low-energy mode, LE) to confirm the presence of antimony.

To obtain better-fit results during the deconvolution of all XRF pixel spectra acquired during scanning, the fitting model implemented in PyMca encompasses the description of various parts. These include features regarding the exciting radiation, the detection system, the specific geometry of our MA-XRF setup and the definition of the sample matrix composition. Within the PyMca model, we incorporate the spectral distribution of primary radiation, corrected for the transmission function of the polycapillary optic. This correction is achieved through a semi-empirical method, which combines a measured spectrum from a scatterer material (a 500 μm thick Mylar foil) with a spectrum simulated using Monte Carlo for the same sample under identical experimental conditions. Additionally, we include an average matrix with a composition similar to the unknown samples under investigation. In the case of pictorial materials, such as the present scenario, a multilayered matrix is employed. This matrix includes gypsum and lead white as preparation layers, totalling a thickness of 120 μm, along with a 30 μm thick layer of linseed oil serving as a binder. Furthermore, the irradiation/detection geometry (including source-sample-detector distance, air path, attenuator thickness and angles) is precisely defined within the model. Lastly, for the fit function of the spectrum, Gaussian peaks with exponential tails, folded with a Gaussian (hypermet function), are utilized, and background is modelled using the SNIP-based method.

## 3. Results and Discussion

### 3.1. Evidence of Naples Yellow

By MA-XRF scanning, it was possible to observe that in some areas of the *Mocking of Christ* lead and antimony are both present (mosaic maps are shown in Figures S6 and S7, respectively), although they do not always correlate. Based on both HE (Figures S1–S8) and LE results (Figure S9), it can be inferred that Naples yellow is used along with lead white to create a dramatic chiaroscuro, usually representing the effect of a single light source such as a candle, as for the typical Utrecht *Caravaggisti*'s style. This can be seen in the faces of all the figures on the left, but most remarkably on Christ's face (Figure 2).

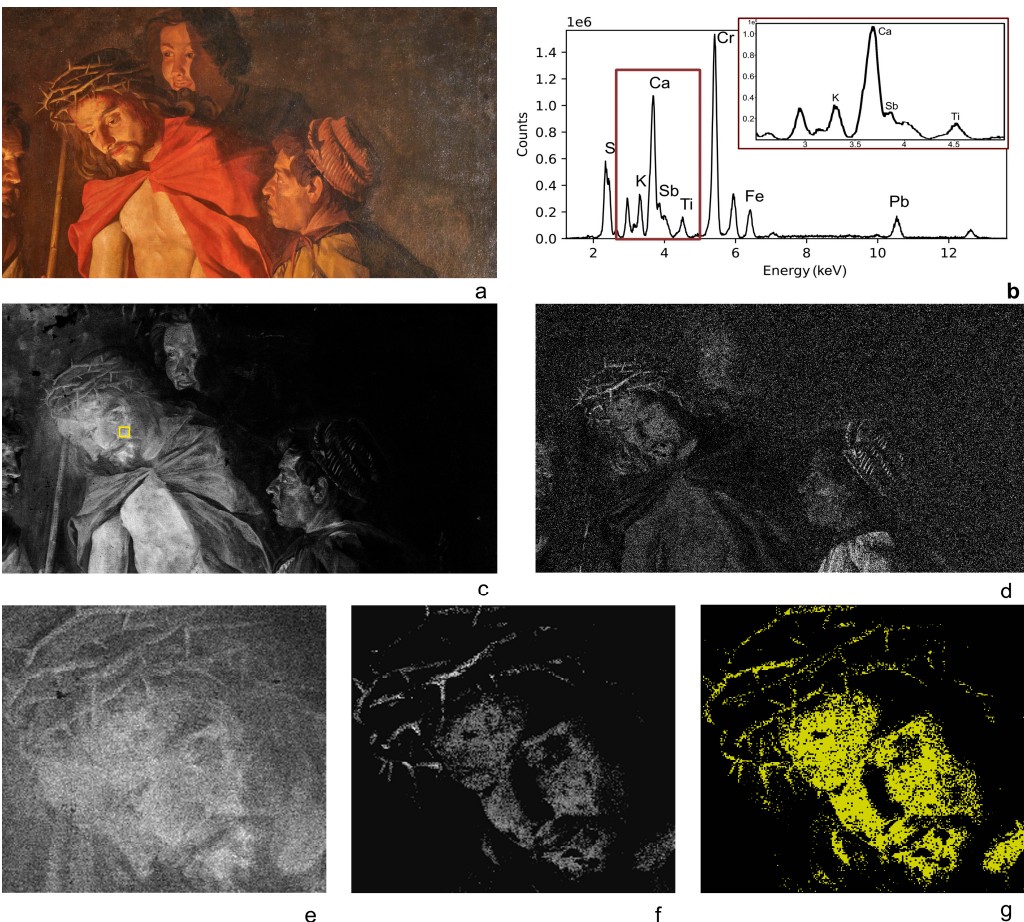

**Figure 2.** VIS-light image (**a**); XRF spectrum collected from an area of Christ's cheek (**b**) from the yellow rectangle in (**c**) using the Cr-anode X-ray tube and comparison of Pb and Sb distribution maps for area P2 of the *Mocking of Christ* obtained with the Rh (**c**,**d**) or Cr (**e**,**f**) target; false-colour image created from the scatterplot of Pb vs. Sb (Figure S10), representing the areas where these elements correlate (**g**).

Analogously, the effects of artificial light on fabrics are expressed using Naples yellow when the candlelight reaches the characters' clothes. While flesh tones are illuminated through lead white (see the areas where Pb and Sb do not correlate in Figure 3b), glowing effects on their clothing are made with Naples yellow over green or light brown-yellow earths, as proved by the deconvoluted spectrum extracted from a region of interest of the rightmost figure's light-green garment (Figure 3c). Similar is Orazio Gentileschi's use of the ternary oxide yellow on the woman's dress of *The Lute Player* between 1612 and 1620 [6].

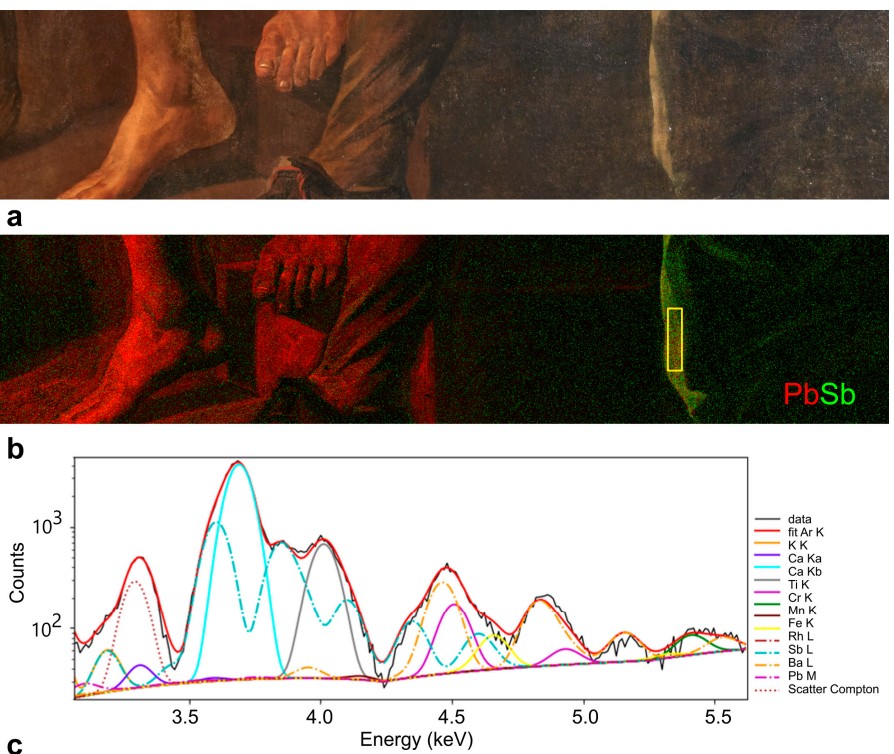

**Figure 3.** Area P6 of the *Mocking of Christ*, VIS-light image (**a**) and false-colour image with Pb distribution shown in red and Sb in green (**b**); deconvoluted contribution from each chemical element (**c**) in a region of interest (ROI, processed with PyMCA software), i.e., the yellow rectangle in (**b**) of the clothing.

### 3.2. Overview of Stomer's Technique and Later Interventions in the Mocking of Christ

Darker red hues and shadows are mostly given by ochres, although cinnabar is also part of the palette. The latter is used for flesh tones, sometimes in very thin layers (see Hg distribution map in Figures 4b and S5), and in the brightest details of the clothing.

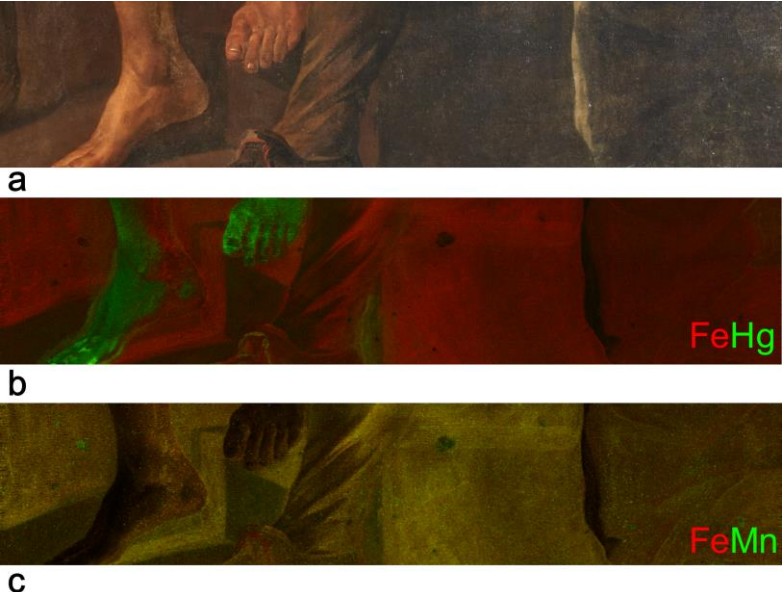

**Figure 4.** Visible light image of a detail in the bottom right area of the *Mocking of Christ* (**a**), with distribution of iron + mercury (**b**) and iron + manganese (**c**).

Sometimes, small red details seem to be painted with a different type of ochre, as observed in the shoelace of the right figure, where Mn is lower or absent, possibly suggesting the use of pure hematite (Figures 4c and S3). In the same area of the painting, K does not seem to be linked to any organic pigment, as it does not show any correlation with aluminium, the major component detected by X-ray analysis for red lakes [25]. In fact, it may represent clay minerals within the brightest green earth used in the right figure's cloths (Figure 5), being potassium distribution consistent with iron presence (see the mosaic maps of K and Fe in Figures S1 and S4, respectively).

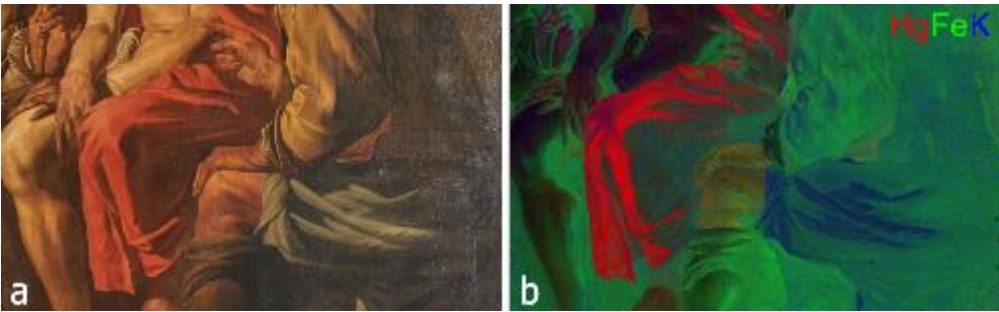

**Figure 5.** Area of acquisition P4, visible light (**a**) and false-colour RGB image (**b**) showing the distribution of mercury (red), iron (green) and potassium (blue).

In the brightest areas of the flesh tones, Stomer's wide use of lead white is evidenced by lead maps from each scanned area. This is consistent with the great diffusion and use of lead white in Western art at that time because of its excellent properties (covering power, durability, lightfastness and rapid drying in oil). The lead map (Figure S6) also showed an important pentimento in the top right area of the painting. The painter's earlier sketch of Christ's face (Figures 2 and 6) seems to show suffering and resignment, as Christ is looking downward, while in the final version the figure gains more dignity, almost directing his glance to the observer. This technical choice seems to account for a compositional maturity, which is evident when we compare this *Mocking of Christ* with earlier versions painted by Stomer in his Roman or Neapolitan period. Two versions—the one at Hôpital St Jean, Brussels [21] and that sold by Sotheby's [26]—show a reversed composition, with Christ on the left and isolated, looking upwards, his face illuminated by an undefined light source. Conversely, the *Christ Crowned with Thorns* in the Chicago Art Market [21] and later in the Norton Simon collection, Pasadena [27], uses the original composition borrowed from his master, with Christ on the right looking downward and a candlelight at the centre of the scene, as van Honthorst had done in his *Mocking of Christ* (ca. 1617) now at LACMA, Los Angeles. However, Stomer goes beyond his master's lesson and plays with lights as much as he plays with materials, using new and established pigments to modify lights and composition, and finally confer boldness and dignity to the main character and the whole scene.

In the maps of lead, signals for this element are also likely to come from the ground, where it correlates with calcium (see Ca distribution in Figure S2), possibly suggesting that the canvas has been prepared with lead white and a Ca-based material like chalk, either in mixture or in different layers. This type of ground is attested in seventeenth-century Italian paintings from different regions and artists, in one or more layers, usually in light to dark colour because of the addition of ochres, minimum or carbon black [28]. Calcium also correlates with strontium in most areas, as expected for materials containing carbonates or sulphates, where the similar atomic ratio of these two elements frequently allows chemical substitution [29].

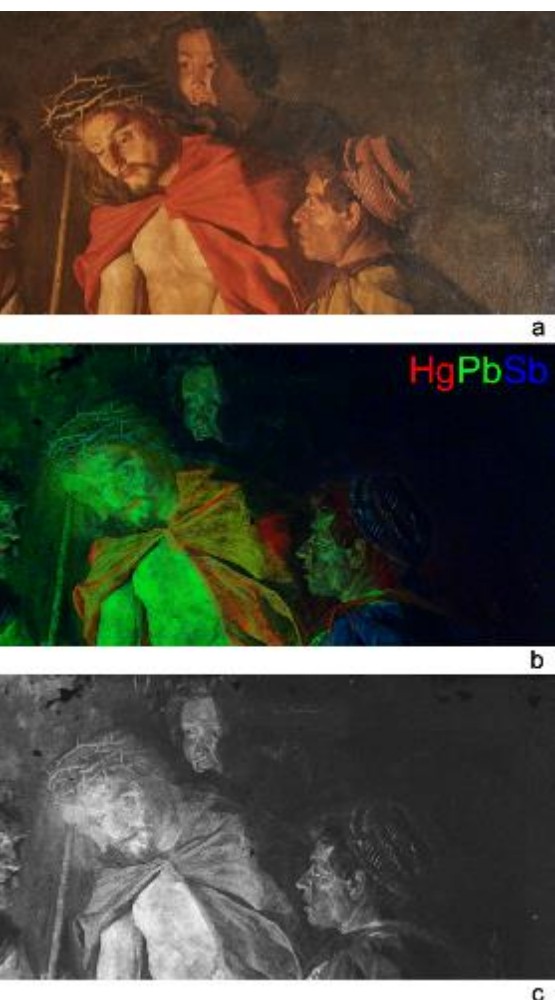

**Figure 6.** Christ's face, visible light (**a**) and false-colour RGB image showing the distribution of mercury (red), lead (green) and antimony (blue) after MA-XRF with the Rh target (**b**); lead distribution is also shown alone in the grey map (**c**).

Modern conservation treatments on the painting can be recognised, especially along the edges of the painting, because of the use of modern pigments introduced in the twentieth century as extenders, usually containing Ba- or Ti-based compounds [30]. In the *Mocking of Christ*, barium white seems to have been used to repair areas of the canvas (see the mosaic map of Ba, Figure S8). The copresence of barium and chromium (map of Cr in Figure S8) in small areas shows that the distribution of these two elements is always consistent and suggests that the paint formulation used for the repairing contains a Ba-based extender and a Cr-based pigment, possibly to give the ground a tone similar to the original one [31].

Although titanium is also in the original ochres (map of Ti in Figure S8), this element sometimes overlaps with zinc (map of Zn in Figure S8), for example in the edge of the top right area of the painting. Their co-distribution does not always match the presence of barium, possibly because they refer to different materials and, hence, different conservation treatments. The use of a modern dark brown pigment containing zinc oxide and titanium dioxide is likely explained as a later intervention with a commercially available pigment, where both elements are due to additives [32,33]. Looking at the distribution of manganese and iron in the areas painted in red, light repainting cannot be excluded, as suggested by a different Mn/Fe ratio.

### 3.3. Naples Yellow in Seventeenth-Century Italy

Regional specificity and experimentation of new raw materials in artists' workshops have been already documented for seventeenth-century paintings [28,34]. Hence, the hypothesis that artists used to collect their sources from pottery workshops is conceivable. From Zalapì [18], we know that in his Neapolitan period (ca. 1633-37, according to the author), not only had Stomer reshaped the lesson learned in Rubens' workshop, but he had also been influenced by the neo-Venetian tradition, which was quite popular in Naples in those years. It is reasonable to think that northern Italian painters not only brought their expertise in style but also in materials and possibly reintroduced pigments from glass-making technology, the leading cities of its production being Venice and Murano in the fifteenth and sixteenth centuries [13]. As a matter of fact, we know from Cipriano Piccolpasso's *Li tre libri dell'arte del Vasaio* (1548) and Vannoccio Biringuccio's *De La Pirotechnia* (1540) that antimony had been reintroduced a century earlier in the manufacturing of yellow pigments for pottery, glass and enamels [8]. Moreover, that small glass objects—the so-called Venetian *paternostri*, yellow or green beads—could give pigments for glass decoration had been reported in the fourteenth century already [10]. Material experimentation might have served Stomer's exploitation of the potentialities of artificial light, already explored in Naples [21], as well as the painter's devotion to Caravaggio and the Caravaggesque artists might have moved him towards the research of new materials to express light. In fact, the good quality of a Neapolitan production of antimonate yellow, which also justifies the epithet of 'Naples yellow', has already been inferred for other painters who resided in Naples in the seventeenth century and particularly for Stomer after 1631 [35,36]. It might not be a coincidence that one of the earliest examples of lead antimonate yellow in Wainwright and co-authors' study [7] has been found in another painting by Stomer, the *Arrest of Christ* (painted in 1630-32), National Gallery of Canada, Ottawa. In 1640, when the painter arrived in Sicily [18,20], antimony-based yellow pigments could have been realistically available there because of a well-developed pottery production [37]. Particularly, the places where he spent his Sicilian period, especially Palermo and Monreale [20], already had a long-standing tradition in the production of maiolica. When Stomer was there, these cities were reaching a high grade of technical and stylistic quality, for example with the works of Girolamo Lazzaro and Andrea Pantaleo [38,39]. The process of contamination between potters' and painters' materials may have been driven by the variety and extent of pottery painters' palettes at the time Italian Renaissance maiolica had become an art form. At the same time, material sharing between artists and artisans could have been fostered by the existence of individual production centres throughout the Italian peninsula with highly specialised features [40]. This could have motivated Lotto's use of Naples yellow in the *Allegory of Chastity* (the so-called *Maiden's Dream* at the National Gallery of Art, Washington, Samuel H. Kress Collection) in 1505–1506 [41], as the painter spent part of 1506 in the Marches, where the first recipes for potters' yellow with Pb and Sb would be published some years later. Analogously, when Lotto completed the *Portrait of Giovanni della Volta with his Wife and Children* in 1547 in Venice [42], he might have had access to the potters' palettes, a few years later than when Biringuccio had described the use of antimony for the manufacture of yellow pigments.

On the other hand, there is evidence that North and South Italy had experienced contaminations in pottery production during the sixteenth century. For instance, Sandalinas has pointed out [12] that the painter Domenego, the most representative Venetian painter of *istoriato* style, was 'significantly influenced by Italian pottery of southern Italy, particularly Sicily and Calabria'.

Therefore, the availability of new materials, specifically from glass making, might have favoured Stomer's mastery of artificial light and the reinterpretation of Caravaggisti's lighting effects slightly earlier than the first recipes for yellows containing Pb, Sn and Sb were collected in the *Ricettario* Darduin of Murano, *Secreti per far lo smalto et vetri colorati* [12]. This is not only consistent with the absence of correlation between tin and antimony or tin and lead in the *Mocking of Christ* but also with the absence of zinc in the original yellow

paint layers. The presence of either element is usually attributed to the use of *tutia*, which cannot be inferred for Stomer's Sicilian production, at least in the paintings analysed in this study. We suggest a geographical explanation for this analytical evidence: *tutia*'s common use in the sixteenth-century Venetian glass and ceramics industry might have been explained as resulting from waves of immigration of eastern Mediterranean glassworkers to Venice, which had not occurred elsewhere. Material evidence on paintings by seventeenth-century artists working in Venice [8] seems to support this hypothesis. For example, for two painters having connections with Venice, the Venetian Giovanni Battista Langetti and Luca Giordano, who spent a year there, Pb-Sn-Sb yellow has been identified: on the late work *Lot and His Daughters* and *Christ's Entry into Jerusalem*, respectively [43]. Within the Castello Ursino collection, the presence of antimonate yellow could not be documented in Stomer's *Tobias healing his father* (ca. 1642) during the same MA-XRF campaign [2]. As this painting comes slightly after the *Mocking of Christ*, it could be inferred that experimentation continued in Stomer's work and led him to use slightly different materials a few years later.

Interestingly, during the same analytical campaign, it was found that there is no antimony in Mattia Preti's *Saint Luke* (or *San Luca nudo, sopra un bove, che sta dipingendo la Madonna*), painted in 1669. It is possible that the painter, active in Naples about 20 years later than Stomer and in good relationship with the Sicilian patron Antonio Ruffo, did not have access to the same materials, as he was already living in Malta when he painted the *San Luke* [44,45].

## 4. Conclusive Remarks

The present study allowed us to establish a continuity in Stomer's production, some of his technological choices and his study of light. MA-XRF imaging documented the use of antimonate yellow, mainly to produce the effects of artificial light on figures' skin and clothing in the *Mocking of Christ* dated to the Sicilian period. Our results are consistent with the palette used by the artist in the *Arrest of Christ*, a painting attributed to the Neapolitan period. Although lead antimonate yellow was not as popular at that time as it would have become a century later, Stomer was already familiar with it and the good quality found in Naples. He continued to use the pigment when he moved to Sicily, probably favoured by a well-established pottery production, which has been described by several authors as the original source of yellow lead pigments. However, because dating and attribution is often uncertain for Stomer's production, it would be difficult to determine the exact chronology of its use.

This study has shown that the presence of Naples yellow in artworks can be determined using MA-XRF imaging. The thorough inspection of spatial correlation in the distribution of lead and antimony constitutes preliminary evidence, which has to be strengthened through the deconvolution of raw spectra extracted from meaningful ROIs. Further investigation may be focused on confirming the presence of lead antimonate yellow in the *Mocking of Christ* and in other paintings by Stomer using Raman spectroscopy, which has proved to be highly diagnostic for this pigment. A systematic analytical campaign on Stomer's production, for example examining paintings from the Neapolitan phase, like the *Adoration of the Magi* in Toulouse, or from the Roman years, such as the *Samson and Delilah* at Galleria Nazionale d'Arte Antica in Rome, might shed light on recurrent practices in his art, as well as on the history of Naples yellow in southern Italy and beyond, before this pigment became so popular one century after the *Mocking of Christ* had been painted.

In the *Mocking of Christ*, MA-XRF imaging also documented the use of other yellow pigments, like ochres, which are also present in their red and brown varieties, cinnabar and the ubiquitous presence of lead white, both in the paint layers and in the ground. Moreover, this investigation shed light on previous conservation treatments, either refilling or retouching with modern products.

**Supplementary Materials:** The following supporting information can be downloaded at https://www.mdpi.com/article/10.3390/heritage7030057/s1, Figure S1: *Mocking of Christ*, mosaic map of K after full scanning by HE MA-XRF (Rh target); Figure S2: *Mocking of Christ*, mosaic map of Ca after full scanning by HE MA-XRF (Rh target); Figure S3: *Mocking of Christ*, mosaic map of Mn after full scanning by HE MA-XRF (Rh target); Figure S4. *Mocking of Christ*, mosaic map of Fe after full scanning by HE MA-XRF (Rh target); Figure S5. *Mocking of Christ*, mosaic map of Hg after full scanning by HE MA-XRF (Rh target); Figure S6. *Mocking of Christ*, mosaic map of Pb after full scanning by HE MA-XRF (Rh target); Figure S7. *Mocking of Christ*, mosaic map of Sb after full scanning by HE MA-XRF (Rh target); Figure S8. *Mocking of Christ*, visible-light image (a) with mosaic maps of the elements attributed to later interventions: Cu (b), Ti (c), Ba (d), Cr (e) and Zn (f) after full scanning by HE MA-XRF (Rh target); Figure S9. Visible-light image of the area scanned with LE MA-XRF (Cr target) and maps of the most representative elements used to paint Christ's face; Figure S10. Scatterplot of Pb vs. Sb, used to build the scatter image of Figure 2 after the analysis by LE MA-XRF (Cr target).

**Author Contributions:** Conceptualization, C.C., F.P.R. and M.B.; Methodology, C.C. and F.P.R.; Validation, C.C. and F.P.R.; Investigation, C.C. and F.P.R.; Formal Analysis, C.C., E.L.R. and M.B.; Writing—Original Draft Preparation, M.B.; Writing—Review and Editing, All Authors; Supervision, F.P.R. All authors have read and agreed to the published version of the manuscript.

**Funding:** This research was funded by the projects H2IOSC—Humanities and Heritage Italian Open Science Cloud (PNRR, Missione 4, Componente 2, Linea di Investimento 3.1 "Infrastrutture di ricerca"); by SAMOTHRACE—Sicilian Micro and Nano Technology Research and Innovation Center (PNRR, Missione 4, Componente 2, Linea di Investimento 1.5 "Ecosistemi per l'Innovazione"); and by the Spoke 5 of the CHANGES project (PNRR, Missione 4, Componente 2, Linea di Investimento 1.3 "Partenariati Estesi").

**Data Availability Statement:** The main data are contained within the article and Supplementary Materials. Additional data presented in this study are available upon request from the corresponding author.

**Acknowledgments:** The authors would like to thank R. Cannavò and Director V. Noto at Museo Civico, Castello Ursino, in Catania for their great support during the analytical campaign and the historical research into Stomer's production, as well as Director Santo Gammino of the LNS-INFN for the infrastructural support. They are also grateful to D. P. Pavone from CNR-ISPC, Catania, for image processing of the Supplementary Materials.

**Conflicts of Interest:** The authors declare no conflicts of interest.

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
