# Peer review of "Naples Yellow Revisited: Insights into Trades and Use in 17th-Century Sicily from the Macro X-ray Fluorescence Scanning of Matthias Stomer’s ‘The Mocking of Christ’"

_heritage, doi:10.3390/heritage7030057_

Round 1

Reviewer 1 Report

Comments and Suggestions for Authors

This work is interesting and well described. The historical research carried out by the authors is a very good complement to the analytical part. The bibliography is complete.

I have only one issue to point out. The authors did not present any molecular evidence of the fact that lead antimoniate was present. Were they able to carry out Raman or visible reflectance measurements? That would have left no doubt on the identification of such pigment. If not, they should justify why, in their opinion, MA-XRF measurements are enough to identify lead antimoniate.

I recommend publication in the present form with some comments to the issue cited above.

Author Response

This work is interesting and well described. The historical research carried out by the authors is a very good complement to the analytical part. The bibliography is complete.

I have only one issue to point out. The authors did not present any molecular evidence of the fact that lead antimoniate was present. Were they able to carry out Raman or visible reflectance measurements? That would have left no doubt on the identification of such pigment. If not, they should justify why, in their opinion, MA-XRF measurements are enough to identify lead antimoniate.

We would like to thank the reviewer for the useful comments and positive feedback.

After a thorough screening of Pb and Sb maps for each scanned area, we further optimised the deconvolution process for spectra extracted from meaningful regions of interest, in order to verify the presence of both elements directly from raw data, as it can be seen in Figure 3c. We clarified this aspect adding a short sentence in the text, and cross-referenced Figure 3c (lines 245-250):

Analogously, the effects of artificial light on fabrics are expressed using Naples yellow, when the candlelight reaches the characters’ clothes. While flesh tones are illuminated through lead white (see the areas where Pb and Sb do not correlate in Figure 3b), glowing effects on their clothing are made with Naples yellow over green or light brown-yellow earths, as proved by the deconvoluted spectrum extracted from a region of interest of the rightmost figure’s light-green garment (Figure 3c).

We are confident this already constitutes clear evidence of the presence of Naples yellow in the Mocking of Christ, although the use of other spectroscopic techniques is desirable as it might provide additional confirmation. We remarked this concept in 4.Conclusive remarks (lines 412-415):

This study has shown that the presence of Naples yellow in artworks can be determined using MA-XRF imaging. The thorough inspection of spatial correlation in the distribution of lead and antimony constitutes preliminary evidence, which has to be strengthened through the deconvolution of raw spectra extracted from meaningful ROIs. Further investigation may be focused on confirming the presence of lead antimonate yellow in the Mocking of Christ and in other paintings by Stomer using Raman spectroscopy, which has proved to be highly diagnostic for this pigment.’

Reviewer 2 Report

Comments and Suggestions for Authors

The paper by Botticelli et al regards an interesting and well conducted MA-XRF study of a large painting by Stomer. The paper is well written and results clearly discussed. Worth to be noted - a choice that I have particularly appreciated, the paper well contextualizes the study into the historical background also adding details on the history of the materials, particular on Naples yellow, main focus of the study. The paper definitely deserves to be publish. I just have little minor remarks that I ask to be addressed.

- ln 50: I know that this is slightly out of the topic, but when listing Pb-based pigments I would mention also massicot for the sake of completeness

- figure S8: instead of naming images with letters, I would name those after the element, as happens in figure S9

- check the captions of figures as sometimes there are missing information (for example, in figure 3 the caption does not describe spectrum c)

- in some map, the merging seems not well finalized both in terms of position and relative normalization. This is particularly visible in Pb map (figure S6)

- this is just a personal observation: sometimes the multi-elemental maps are not that easy to read, mainly for the choice of colours (dark blue does not have high contrast for example). Personally, I find much more efficient bw maps, but I would suggest to change the colours in some cases. For example, in figure 6b Sb in barely visible.

Author Response

The paper by Botticelli et al regards an interesting and well conducted MA-XRF study of a large painting by Stomer. The paper is well written and results clearly discussed. Worth to be noted - a choice that I have particularly appreciated, the paper well contextualizes the study into the historical background also adding details on the history of the materials, particular on Naples yellow, main focus of the study. The paper definitely deserves to be published. I just have little minor remarks that I ask to be addressed.

The authors would like to thank the reviewer for the positive feedback. We have addressed comments at our best. Details below:

- ln 50: I know that this is slightly out of the topic, but when listing Pb-based pigments I would mention also massicot for the sake of completeness

We thank the reviewer for this suggestion, which makes the introduction more complete. We have included a short sentence which mentions massicot among the Pb-based pigments and further discusses the misleading terminology (lines 58-62):

It has been also reported that ‘A comparable recipe producing a lead tin-based yellow under the heading ‘massicot’ is to be found in a sixteenth century Flemish text, confirming the equivalence of the two terms’ [5], although the term massicot more commonly describes an orthorhombic lead oxide, PbO, produced by heating lead white and used as yellow pigment.

This action has implied the inclusion of a new reference, now [5], which is also included in the current list of references as follows:

  1. Eastaugh, N.; Walsh, V.; Chaplin, T.; Siddall, R. Pigment Compendium - A Dictionary and Optical Microscopy of Historical Pigments; Elsevier Ltd.: Oxford, 2016; ISBN 9780333227794.

- figure S8: instead of naming images with letters, I would name those after the element, as happens in figure S9

Figure S8 has been relabelled as suggested.

- check the captions of figures as sometimes there are missing information (for example, in figure 3 the caption does not describe spectrum c)

We checked all the captions and modified the following:

Figure 2. VIS-light image (a), XRF spectrum collected from an area of Christ’s cheek (b, from the yellow rectangle in c) using the Cr-anode X-ray tube and comparison of Pb and Sb distribution maps for area P2 of the Mocking of Christ obtained with the Rh (c, d) or Cr (e, f) target; false-colour image created from the scatterplot of Pb vs Sb (Figure S10), representing the areas where these elements correlate (g).

Figure 3. VIS-light image (a) and false-colour image (b, Pb distribution shown in red, Sb in green) of area P6 of the Mocking of Christ, with deconvoluted contribution from each chemical element (c) in a region of interest (ROI processed with PyMCA software, yellow rectangle in b) of the clothing.

Figure 6. Christ’s face, visible light (a) and false-colour RGB (b) image showing the distribution of mercury (red), lead (green) and antimony (blue) after MA-XRF with the Rh target; lead distribution is also shown alone in the grey map (c).

- in some map, the merging seems not well finalized both in terms of position and relative normalization. This is particularly visible in Pb map (figure S6)

We agreed with the reviewer and modified Figure S6, as well as the other Supplementary figures of mosaic maps (S1-S7) with optimised stitching.

- this is just a personal observation: sometimes the multi-elemental maps are not that easy to read, mainly for the choice of colours (dark blue does not have high contrast for example). Personally, I find much more efficient bw maps, but I would suggest to change the colours in some cases. For example, in figure 6b Sb in barely visible.

Figure 6b is a RGB image. By definition, it comes as a false-colour image where the distribution of three selected elements is represented in red, green and blue. We think it is not correct to replace blue with a different colour in this figure. However, we modified Figure 4, which now represents the distribution of Hg (b) and Mn (c) in green.

Reviewer 3 Report

Comments and Suggestions for Authors

Dear authors,

I would like to congratulate them on the work they did.

- I would like to suggest that the authors improve the description of figure 2. The description is a little confusing.

- I would like to suggest to the authors that the conditions used to build the FIT model in Pymca.

Author Response

I would like to congratulate them on the work they did.

We would like to thank the reviewer for the useful comments and positive feedback.

 - I would like to suggest that the authors improve the description of figure 2. The description is a little confusing.

 We modified the description of Figure 2 as follows (lines 240-244):

‘Figure 2. VIS-light image (a), XRF spectrum collected from an area of Christ’s cheek (b, from the yellow rectangle in c) using the Cr-anode X-ray tube and comparison of Pb and Sb distribution maps for area P2 of the Mocking of Christ obtained with the Rh (c, d) or Cr (e, f) target; false-colour image created from the scatterplot of Pb vs Sb (Figure S10), representing the areas where these elements correlate (g).’

- I would like to suggest to the authors that the conditions used to build the FIT model in Pymca.

The conditions used to build the FIT model in Pymca are the following (now described in 2.2 Macro-XRF scanning, lines 212-229):

To obtain better fit results during the deconvolution of all XRF pixel spectra acquired during scanning, the fitting model implemented in PyMca encompasses the description of various parts. These include features regarding the exciting radiation, the detection system, the specific geometry of our MA-XRF setup, and the definition of the sample matrix composition.

Within the PyMca model, we incorporate the spectral distribution of primary radiation, corrected for the transmission function of the polycapillary optic. This correction is achieved through a semi-empirical method, which combines a measured spectrum from a scatterer material (a 500μm thick Mylar foil) with a spectrum simulated using Monte Carlo for the same sample under the identical experimental conditions.

Additionally, we include an average matrix with a composition similar to the unknown samples under investigation. In the case of pictorial materials, such as the present scenario, a multilayered matrix is employed. This matrix includes gypsum and lead white as preparation layers, totalling a thickness of 120μm, along with a 30μm-thick layer of linseed oil serving as a binder. Furthermore, the irradiation/detection geometry (including source-sample-detector distance, air-path, attenuators thickness, and angles) is precisely defined within the model. Lastly, for the fit function of the spectrum, Gaussian peaks with exponential tails, folded with a Gaussian (hypermet function), is utilized and background is modelled using the SNIP-based method’.

The construction of the model described above necessitates an extensive characterization of the instrumental setup employed during the measurements. However, in cases where such characterization is lacking, it is still feasible to conduct the spectrum fit procedure by meticulously defining the peaks fit function and the background subtraction function.
